# Towards Ecological Management and Sustainable Urban Planning in Seoul, South Korea: Mapping Wild Pollinator Habitat Preferences and Corridors Using Citizen Science Data

**DOI:** 10.3390/ani12111469

**Published:** 2022-06-06

**Authors:** Hortense Serret, Desiree Andersen, Nicolas Deguines, Céline Clauzel, Wan-Hyeok Park, Yikweon Jang

**Affiliations:** 1Division of EcoScience and Department of Life Sciences, Ewha Womans University, Seoul 03760, Korea; desireeka93@hotmail.com; 2EBI, CNRS, Université de Poitiers, 86000 Poitiers, France; nicolas.deguines@univ-poitiers.fr; 3LADYSS UMR 7533 CNRS, Université Paris Cité, 75013 Paris, France; celine.clauzel@univ-paris-diderot.fr; 4Department of Forest Resources, Kookmin University, Seoul 02707, Korea; robinaus@kookmin.ac.kr; 5Interdisciplinary Program of EcoCreative, Ewha Womans University, Seoul 03760, Korea

**Keywords:** pollinators, citizen science, multi-family habitats, ecological corridors, urban landscapes, conservation, species distribution modeling, graph modeling

## Abstract

**Simple Summary:**

Urban landscapes, though a primary contributor to habitat fragmentation, have the potential to facilitate habitat connectivity for native pollinator species, especially with strategic green space planning. Citizen science data have been proven to be useful to address conservation issues in urban areas, especially where knowledge is lacking about species richness and habitat preferences. In this study, we used data collected by a citizen science program between 2016 and 2018 to determine which families of pollinators were the most frequently observed in Seoul, with which habitats pollinators had the highest affinities, and what parts of the city facilitated habitat connectivity. We found that the most suitable habitats for multiple families were located in public parks, university campuses, and Cultural Heritage sites. These results are directly applicable in order to better understand urban planning stakes for pollinators and wildlife in general and provide avenues for improvement to recreate functional greenways in a dense city like Seoul.

**Abstract:**

The preservation and restoration of habitats and ecological connectivity inside cities is crucial to ensure wildlife can find suitable areas to forage, rest and reproduce, as well as to disperse, thereby allowing metapopulation functioning. In this study, we used data collected by a citizen science program between 2016 and 2018 to determine which families of pollinators were the most frequently observed in Seoul and with which habitats pollinators had the highest affinities. Using species distribution modeling and landscape graph approaches, we located the main habitats and corridors to reinforce connectivity for six pollinator families. Finally, we identified habitats and corridors where conservation actions should be prioritized. In total, 178 species belonging to 128 genera and 60 families were observed. Hymenopterans were the most recorded, followed by dipterans and lepidopterans. The most suitable habitats for pollinators were constituted of public parks, university campuses, and Cultural Heritage sites. In a dense city like Seoul, most of the conservation corridors are located in built-up areas. Innovative urban planning and architecture are therefore required as well as the setting-up of ecological management practices to lead to a more sustainable urbanism for pollinators and wildlife in general.

## 1. Introduction

As the world continues to urbanize, the conciliation between urban growth and the preservation of functional ecosystems in cities appears to be one of the main challenges of the 21st century [1]. The more cities are sprawling, the more natural habitats shrink and become fragmented, leading to biodiversity erosion and homogenization [2,3]. The preservation and restoration of ecological connectivity inside cities is therefore crucial to ensure sustainable habitats for wildlife for foraging, resting, and reproduction as well as corridors enabling dispersal [4]. 

Nature-based solutions inside cities are vital for the well-being and health of city-dwellers, and for adaptation to climate change. Recent works showed that besides the numerous ecosystem services urban green spaces provide, such as reduction of urban heat island effects [5] or flood control [6], accessibility to urban green spaces also brings benefits to physical health [7,8]. Daily contact with nature where people live and work may improve mental health. Following the works initiated by Kaplan [9] describing the restorative effects of nature, numerous studies showed that access to green spaces and contact with nature was associated with a decrease in stress or anxiety and better emotional well-being [10,11,12]. 

Seoul experienced explosive population growth (from 1.6 million in 1955 to 10.6 million in 1990) that led to an uncontrolled urban sprawl [13]. In accordance with the need for urban planning regulations, urban containment policies were implemented during the 1970s and the 1980s, and strategic development plans emerged during the 1990s. More recently, the Seoul Metropolitan Government engaged itself in large urban regeneration projects to improve the quality of life for Seoulites (https://world.seoul.go.kr/urban-green-space-to-be-connected-and-extended-by-2000-km-network-of-green-paths/ accessed on 23 February 2022). In 2004, the Cheonggyecheon Restoration Project, the most expensive river restoration in the world thus far, took place in Seoul and resulted in the removal of an urban highway. The original stream was re-opened, and a 6 km-long continuum of green spaces along the stream was created. This project illustrates the city’s effort to create more eco-friendly and people-oriented urban spaces. The green spaces created by this project are conciliating city-dwellers’ well-being and biodiversity conservation by locally improving air quality, reducing the urban heat island effect locally [14], and creating suitable habitats for numerous groups of species [15]. 

Among species present in urban areas, wild pollinators are important for maintaining urban ecosystems and can be considered bioindicators of environmental health [16,17]. Pollination is indeed one of the most important ecosystem services and is crucial for the persistence of diverse plant communities [18] and to ensure the productivity of urban agriculture [19]. Researchers have only recently realized that cities could be designed as “refuges” for wild pollinators thanks to areas creating suitable and continuous habitats such as parks, residential gardens, or green spaces at business sites. Indeed, some of these urban green spaces are hosting melliferous flowers, grasslands, hedges, and other favorable micro-habitats for pollinators, creating a network of nesting areas within cities [20,21]. To this end, there is a need to better understand the ecological requirements of pollinator communities within cities, highlighting habitat preferences and identifying potential corridors in the city as the first steps to implementing conservation actions [22]. 

Biodiversity conservation is a societal issue involving the general public and the development of citizen science is an effective way to monitor biodiversity at a large spatiotemporal scale [23] engaging the general public in scientific research. Many citizen science programs monitoring pollinators were launched recently [24,25], tied to the growing awareness of pollinator decline [26]. Data collected by volunteers have already been successfully used to better understand pollinators’ landscape affinity in France [24] and the distribution of particular groups such as six bumblebee species in Japan [25]. Recently, some researchers have been able to map the distribution of several UK social wasps in only two weeks thanks to a citizen science initiative [27].

Many citizen science programs use presence-only data, which is the easiest way to collect information about species presence. Opportunistic data (as can be produced with some citizen science projects) may be used to estimate distribution and abundance trends [28,29,30,31] and species responses to management practices [32]. Such information is critical to help decision-makers define actions for conservation. 

In 2016, a citizen science project focusing on pollinators was started in partnership with a Korean publishing company (Donga Science, Seoul, Korea). This project is part of a larger program, called “Earth Loving Explorers” (http://kids.dongascience.com/earth/post accessed on 23 February 2022), the first initiative of citizen science in South Korea, to address research questions in ecology and conservation biology. We used data collected by volunteers between 2016 and 2018 [33] to first describe the composition of the pollinators’ community in Seoul and its richness by land-use types. Secondly, we used species distribution models (SDM) to characterize the habitat suitability of six major pollinator families in Seoul. This analysis led us to the identification of the areas constituting the best habitats from a multi-family point of view. Finally, we determined, using a graph-modelling approach and the calculation of connectivity metrics, the best locations to maintain existing corridors contributing to connectivity and to restore new ones at the scale of Seoul. 

## 2. Materials and Methods

### 2.1. Study Area

Although the pollinator citizen science project is open to the whole country, our analysis focused on Seoul, the capital of the Republic of Korea, because most of the observations were done in this area. Seoul spans approximately 38 km from north to south and 45 km from east to west (Figure 1). As one of the key global cities in Asia today and the economic heart of the country, it hosts 22% of the country’s firms and 20.6% of total employment [34]. Approximately ten million people are living in Seoul, making it one of the densest cities in the world with 16,000 people/km². Including the surrounding areas, the Seoul Metropolitan area holds 25 million people, half of the entire Korean population [34].

### 2.2. Data Collection

Data were collected from 2016 to 2018 using a citizen science project monitoring pollinators with a standard protocol based on photography [33]. For this study, 1897 pictures from 145 observers were used (Figure 2).

A cellphone application dedicated to this program was developed to collect data from volunteers who were asked to take pictures of any insects landing on flowering plants. Along with photographs, the application allowed recording of the date, time of day, GPS coordinates, and environmental conditions. GPS locations were registered at the moment where observations began and were not modified if the uploading was completed later. Pictures without GPS location, containing insects that were not on flowers or too blurry for identification were discarded. Identification of viable pictures at the species-, genus- or family-level, according to picture quality and the possibility of identification, was conducted by professional researchers, scientific partners of the company, and amateur entomologists. They were validated by an entomology expert, Dr. Lee Heung-Sik from Plant Quarantine Technology Center. 

When the program was launched in 2016, participants were only asked to send pictures without any constraint of observation effort. However, in 2017 a protocol was established in order to get a more structured dataset. Participants were asked to do observations on the same plant and to take pictures of all the insects landing on it for 15 min. However, participants were suspected to have misunderstood the protocol as most of the observations contained only one species, as described in Serret et al. [33]. Therefore, we chose to consider all the data as presence-only data and used the pictures from 2016 to 2018. 

### 2.3. Estimating Pollinator Richness

We first described the proportions of the taxonomic diversity of pollinators and the most frequent species observed in Seoul. This was compared with previous studies to evaluate if our dataset was representative of species composition. 

We relied on land-use data from 2017 with a resolution of 10 m, provided by the Ministry of the Environment (https://egis.me.go.kr/ accessed on 23 February 2022). Land-use was categorized into nine classes: agriculture, forest, industrial and commercial areas, meadows and grasslands (including planted lawns, planted or natural meadows), open and mineral areas, public and cultural facilities, residential areas (mainly residential towers with small green spaces or lower buildings), transportation, wetlands and water (Figure 1). 

As each land use was not sampled with the same effort and because observed species richness is highly dependent on it, we calculated the estimated species richness, which is based on extrapolation of the species accumulation curves and allows the comparison among areas with different sampling efforts [35]. According to Palmer [35], “*the jackknife and the bootstrap are nonparametric resampling methods for obtaining estimates*”. We, therefore, calculated the jackknife and bootstrap indices with the function “specpool” from the ‘vegan’ package [36] in R ver. 3.6.1. 

Pollinator richness was estimated according to the number of genera observed per land-use class. Indeed, species-level identification was sometimes not possible. We first performed a resampling based on geographic location. As our data were based on participants’ observations, they did not follow a balanced observation pressure for every land class. Furthermore, some participants did a lot of observations in the same area, leading to the creation of pseudo-replicates (Figure 2). To correct the replication of observation within a small area, a fishnet of 250 m square was created on ArcGIS version 10.5.1 (ESRI, Redlands, CA, USA) and all the redundant observations from each genus located within the same square were removed. 

### 2.4. Land-Use Affinity of Major Pollinator Families

We chose to consider the six main families best representing the communities of wild pollinators observed in Seoul: Apidae (181 occurrences), Syrphidae (225 occurrences), Megachilidae (37 occurrences), Halictidae (147 occurrences), Lycaenidae (74 occurrences), and Pieridae (142 occurrences). The Apidae observation excluded *Apis mellifera* and *Apis cerana* because their populations in Seoul are not feral but linked to beehive locations. 

To assess the land-use preferences of these six families, we calculated an affinity index as described by Deguines et al. [24]. As was the case for that study, our sampling was not spatially well balanced. To control this heterogeneity, we had to characterize the land-use proportions surrounding each observation relative to all the other ones performed in Seoul. We first calculated, for each observation, the proportion of each land-use class in a 100 m radius. We deliberately chose a small range due to the highly heterogeneous landscape of Seoul. For each collection’s land-use class, we subtracted this value from the proportion of this land-use class among all the observations, according to this equation: (1)$${LU}_{o,l}=P_{o,l}-P_{all,l}$$
where *P_o,l_* and *P_all,l_* are, respectively, the proportions of the land-use class *l* within 100 m of the observation *o* and the proportions of the land-use class *l* within 100 m of all the observations in Seoul. *LU_o,l_* is then the relative land-use type *l* index of an observation *o*. 

We used the package ‘ggplot2′ [37] in R ver. 3.6.1 to create, for each family, the boxplot illustrating the range of relative land-use class proportions.

### 2.5. Habitat Suitability Modeling

Species distribution models (SDMs) are often used to map the geographical distribution of species or to highlight the most suitable areas for the survival of a species of conservation concern [38]. Maximum entropy (MaxEnt [39]) modeling is one such method for predicting suitability based on a species’ response to each environmental variable compared to background conditions, and this technique is more and more popular with the amount of presence-only data gathered with citizen science projects [32,40,41]. In the current study, we wanted to have a community approach and chose to use this modeling at the taxonomic family level. 

#### 2.5.1. Variables

For this analysis, we focused on the 6 main families of wild pollinators: Apidae, Megachilidae, Halictidae, Pieridae, Lycaenidae, and Syrphidae. A total of 52 variables were used in building models for testing in MaxEnt 3.4.1. These included landscape variables, elevation (DEM, USGS; data provided by the Water Resource Management Information System), slope (DEM, USGS), aspect (DEM USGS), distance from hives (as pollinators and honey bees may be in competition in the urban area as mentioned by Ropars et al. [42]), normalized difference vegetation index (NDVI, Landsat 8), and population density in 2015 (data provided by the Ministry of the Interior and Safety)(total seven variables). For landscape variables, we derived Euclidean distance from each of the nine landscape classes (a total of nine variables). We also calculated the percent cover of each class within 100 m, 200 m, 500 m, and 1 km using focal statistics (a total of 36 variables; four for each of nine classes). All models were built using rasters of environmental variables at a resolution of 20 m.

#### 2.5.2. Maximum Entropy

We used the R package “MaxentVariableSelection” [43] to test MaxEnt models for the six pollinator families [44]. This package runs MaxEnt models for multiple runs, removing variables that are highly correlated and do not contribute greatly to the models. This package can be run with different beta multiplier values representing a “smoothing” of a species’ response to each model variable. Initial models contained all 52 variables, and variables were retained if the Pearson’s correlation coefficient between variables was <0.8 and if the contribution was >2. We used beta multipliers of 1, 2, and 3 to test different levels of variable response smoothing. Selected models were those with the lowest AICc (Akaike information criterion corrected for sample size). All selected models were rerun in the MaxEnt platform with a 20% random test percentage for ten bootstrap replicates set to random seed and duplicate presence records removed. Final models were thresholded for suitability/non-suitability at the maximum TSS (true skill statistic) threshold for predefined MaxEnt thresholds (Table 1).

#### 2.5.3. Classification

Models were classified into five classes corresponding to five levels of suitability using the following thresholds: Minimum training presence, 10 percentile training presence, Equal training sensitivity and specificity, and Equate entropy of thresholded and original distributions. 

### 2.6. Multi-Family Habitats and Corridors

This analysis was conducted to identify the location of the most suitable habitats and the strategic corridors linking them from a multi-family point of view. We used the five classes corresponding to the five levels of suitability (see Section 2.5) and considered, for each family, the highest suitability class as “Best Suitable Habitat” (BSH). After having created a BSH layer for each family, we used the “Weighted Sum” tool (Spatial Analyst Toolbox) of ArcGIS version 10.5.1 (ESRI, Redlands, CA, USA), superposed the six BSH layers (one per family), and created a map illustrating a range of “Multi-families habitats” going from 0 (area not suitable for any of the families), to 6 (common BSH areas for the 6 families). 

For each pollinator family, we modeled the corridors linking the BSH. We used Graphab [45], a graph-based landscape modeling software [46]. A graph is composed of nodes representing the most suitable habitats for a targeted group of species and links the displacement between them, according to the targeted group of species’ movement abilities across this landscape. For each family, the nodes were defined by the BSH. Links between patches were generated using least-cost paths, as this method is the most appropriate to model the movements of species in a highly heterogeneous landscape [46]. Resistance values were attributed according to the five classes of suitability given by Maxent analysis ranging from 1 (the least resistance value) for the BSH, (i.e, habitat nodes) to 10,000 (the most resistant value, lowest values of suitability). Then, corridors between habitats were created using the “Corridor” function of Graphab (https://sourcesup.renater.fr/graphab/download/manual-2.4-en.pdf accessed on 23 February 2022). These corridors represent the set of possible paths connecting two patches and having a distance lower than the dispersal distance capacity, set to dispersal capacities of each family found in previous studies (Appendix A). Eventually, we obtained a set of nodes with a set of corridors for each family. As previously, we used the tool “Weighted Sum” from Spatial Analyst Toolbox of ArcGIS version 10.5.1 (ESRI, Redlands, CA, USA) to superpose the six layers of corridors.

### 2.7. Map Synthesis for Sustainable Urban Planning and Conservation Strategies

From the multi-family habitat and corridor analyses, we selected the areas suitable for at least four families, i.e., for which at least four BSH layers were superposed. For more clarity, we created a map combining these multi-family habitats and multi-family corridors. We identified on the map the names of the places (mainly public parks, university campus, and Cultural Heritage sites) corresponding to the reservoir habitats. 

### 2.8. Corridor Prioritization for Action

To maintain or enhance connectivity for pollinators in Seoul, we identified “conservation corridors”, i.e., corridors that must be maintained and reinforced because of their high contribution to current connectivity. We also identified by visual analysis some “Potential restoration corridors”, i.e., where restoration actions could be done between habitat patches with moderate to high contribution to connecting patches that are not connected yet. This approach examines the potential restoration corridors and does not take into account economic, social nor political issues associated with restoration. 

To prioritize the corridor, we created a graph for which we considered the reservoir habitats (See Section 2.7) as patches. Resistance costs were ranked according to the number of superposed BSH per area. The highest resistance cost, 1000, was attributed to areas suitable for no or only one family. Then, a cost of 100 was attributed to areas suitable for two families, 10 for three families, and 1 for the reservoir habitats. Then, four graphs were generated according to four dispersal distances corresponding to the average dispersal capacities of the species belonging to each family: 200 m (Halictidae), 500 m (Lycaenidae), 1 km (Megachilidae, Syrphidae, Apidae), and 3 km (Pieridae).

To assess the contribution of links and patches to connectivity, we used the Probability of Connectivity index (PC) developed by Saura and Pascual-Hortal [47], defined as ‘*the probability that two animals randomly placed within the landscape fall into habitat areas that are reachable from each other*’ [47]. The PC was calculated as follows:(2)$$PC=\frac{\sum_{i=1}^{n}\sum_{j}^{n}a_{i}a_{j}p_{ij}}{A²}$$
where *n* is the total number of patches, *a_i_* and *a_j_* are the areas of patches *i* and *j*, respectively. *p_ij_* is the probability that *i* and *j* are connected by *a* link and *A* is the surface area of the entire graph. An exponential function can be used to calculate *p_ij_* as follows:(3)$$p_{ij=e^{-kdij}}$$
where *d_ij_* is the least-cost distance between *i* and *j*, and k (0 < *k* < 1) expresses the reduction in dispersal probabilities resulting from this exponential function [47]. These metrics were calculated for both patches and links for each dispersal distance. 

To identify the “Conservation corridors”, we selected, for each dispersion capacity, 30% of the links with the highest PC values and superposed them. To synthesize the values of the patches for connectivity, we calculated the average PC value for each patch and dispersion distance and ranked them from low to high connectivity potential. The “Potential restoration corridors” correspond to links that could recreate continuity between patches with moderate to high connectivity potential. These results were synthesized on a map. 

## 3. Results

### 3.1. Community Composition

In total, 178 species belonging to 128 genera and 60 families were observed before the 250 m fishnet was applied (Table 2). Hymenopterans were the most recorded, representing 57.6% of all observations. Dipterans and Lepidopterans were the other major orders observed, accounting for, respectively, 17.7% and 15.6% of the observations (Table 2). 

Among the wild pollinators (i.e., *Apis mellifera* and *Apis cerana* excluded), *Pieris* was the most common genus (126 observations, 11.8% of observations), followed by the genus *Bombus* (107 observations of *Bombus ignitus* and *Bombus ardens*, 10%), *Halictus* (80 observations, 7.5%), *Lasioglossum* (56 observations, 5.3%) and *Eristalis* (50 observations of *E. arbustorum* and *E. cerealis* mainly). *Episyrphus balteatus* was one of the most common species observed with 64 observations as well as *Pseudozizeeria maha* (46 observations), *Sphaerophoria menthastri* (38 observations), *Nysius plebejus* (37 observations), and *Xylocopa appendiculata* (30 observations). Ninety species (50% of the species observed) were observed only once. Among the 128 genera, the 19 most frequent ones accounted for more than 75% of all observations. 

### 3.2. Richness and Land-Use Affinity by Family

According to the calculation of the extrapolated index, the richness was highest in meadows with 61 different genera observed (48% of the total genera observed), followed by the agriculture areas (53 genera), the residential areas (52 genera), and the forests (44 genera) (Table 3). 

Regarding the land-use affinity of each family (Figure 3), most of them, particularly, Pieridae, Apidae, and Halictidae, showed an affinity for agricultural areas. On the other hand, forest areas appeared to be the ones with which most of the families had fewer affinities. It was particularly true for both of the Lepidoptera families, the Pieridae and Lycaenidae. Most of the families were also showing positive affinities with the open areas and more particularly the Apidae, Megachilidae, and the Halictidae families. 

### 3.3. Suitable Habitat by Family

High-probability areas of pollinator occurrence were heterogeneous among different families (Figure 4), but the historical center of Seoul, with parks around the palace (see Figure 5, C1) and Namsan Park (see Figure 5, P4), rich in forest and green spaces, were areas of high-probability distribution for all families. Yeouido Island (see Figure 5, P10), rich in open areas and meadows as well as open areas along the Han River, contains suitable habitats for almost all families. 

### 3.4. Multi-Family Habitats and Corridors

Main habitats (Figure 5, orange to dark red) represent 2145 ha in Seoul or 3.7% of the city area. They are slightly more numerous in the north than on the south side of the Han River. Two Cultural Heritage Sites corresponding to historical palaces (in the northern part of Seoul) and royal tombs preserved in a forested park (southeast part of Seoul) also constituted some of the multi-family habitats.

Most of the main habitats are located in Seoul’s main public parks (see Figure 5, Public Parks) such as the Olympic Park (P1), Children’s Grand Park (P2), Yeouido Park (P10), or the Worldcup Park (P12). Large university campuses like Yonsei University (U1), Konkuk University (U7), and Korea University (U2) constitute suitable habitats as well as parks belonging to Heritage Culture Sites such as the Gyeonbokgung and Changdeokgung Palaces (C1), in the north of the city and the Seongjeongneung Royals Tombs, in the south-east (C2). Some elementary, middle and high schools are also part of the urban structures which could potentially benefit multiple pollinators’ families. Most of the time, these infrastructures contain planted open areas and are close to forests and parks, making them potentially suitable habitats for flower visitors. 

The main corridors (Figure 5, light to dark green) represent 16,212 ha of the city or 28.3% of the entire city area. The map of Figure 5 also illustrates the isolated patches, such as C2 (south-east) or P8 (south-west), and the areas lacking suitable habitats or corridors. 

### 3.5. Prioritizing Habitats and Corridors for Conservation

The location of the multi-family habitats and their connectivity potential (Figure 6) ranged from low (blue patches) to moderate (orange patches) and high (red patches). The patches having the highest connectivity potential are located in the northern part of the city. One of them is located in the southeast part of the city, the Olympic Park (also see Figure 5, P1). 

The conservation corridors are mostly located around the patches having high and moderate connectivity potential. Most of them are also located in the northern part of the city, connecting the patches with high connectivity potential. Potential restoration corridors were overlaid to identify areas where restoration actions could be done in order to reconnect patches. Several studies showed how habitat restoration such as recreation of ecological meadows, invasive plant removal, or adapted seed mixes had benefited wild bees’ richness and abundance [48,49]. Tonietto et al. [49] more specifically showed how mowing frequency was negatively impacting wild bees’ richness and abundance, suggesting that reducing mowing frequency should also be considered in restoration actions. The Han River and Eastern Seoul appear to be potential areas where such restoration actions could be done.

## 4. Discussion

### 4.1. Enhancing Scientific Knowledge about Pollinators in Seoul with Citizen Science

Thanks to the pollinator data collected by citizen scientists between 2016 and 2018, we were able to identify a total of 178 wild pollinator species. As mentioned before, we assumed that there were more species, but picture identification imposed some limits and some of the insects could be identified only to the family or genus level. 

For the city of Seoul, our study is the first to investigate the diversity of pollinators. Our study showed that the most represented family was the Hymenopterans, meaning that the pollination service within Seoul is more dependent on this type of family, followed by Dipterans and Lepidopterans. Knowing that fact, it is possible to adapt conservation and ecological restoration measures targeting these families of pollinators. 

A literature review of the species of pollinators observed in agricultural crops and wild flowering areas in South Korea [50] may be used to assess the reliability of our results. Our sampling contained half of the species they listed, and the most frequent species found in Choi and Jung [50] were similar to the ones in our sampling such as *Apis mellifera*, *Eristalis cerealis*, *Xylocopa appendiculata*, *Pieris rapae* or *Episyrphus balteatus*. 

The results based on land-use affinities showed that agricultural areas were favorable to all the groups of pollinators, showing that these areas were a way to create suitable habitats for pollinators while addressing provisioning services in the city of Seoul. Transportation areas may also be a way to create habitats for pollinators and wildlife in general. Indeed, the surroundings of the tracks are often less managed and are presenting small patches of wastelands with various species of plants. All of these areas could be better considered in order to create a network of suitable habitats for pollinators. 

As is the case for most citizen science programs, the sampling was affected by the location where people took pictures. Due to a biased sampling design, we also lacked sufficient information from specific types of habitats such as forests, private backyards that are not common in a city like Seoul, and restrictive access areas, such as the Yongsan Military Base. Even if these issues can nevertheless be addressed by statistical solutions [27,51], more data would be needed to strengthen the results and predictions of this work, and further field campaigns (or call to observers) targeting under-represented habitats could address current sampling limitations. 

### 4.2. Improving Green Infrastructure for Pollinators in Seoul

The main interest of our study was to identify the most strategic areas for pollinator conservation in Seoul and to encourage the stakeholders to set up conservation actions. Taking into account the sampling bias, our study showed that open meadows (found mainly in public parks), agriculture areas, and residential areas were the best to host diverse communities of pollinators. Other recent studies showed that in British cities, the pollinator ‘hotspots’ were comprised of residential gardens and community gardens [52]. 

Most of the public parks are managed by the City of Seoul, making it the principal actor that could be involved in setting up policy toward ecological management of green spaces. However, no clear management practices are set in Seoul yet. Comparing our study to current management practices could be a first step to assessing the relevance of current practices with ecological objectives. This study provides strategical information to target areas where ecological management practices should be increased and implemented in order to improve their capacity for hosting diverse populations of pollinators. The patches identified with “High connectivity potential” (Figure 6) could be targeted as priority areas to implement conservation and management actions. 

Conservation and ecological management actions to enhance habitats for pollinators could be accomplished at different levels: conservation, restoration, and creation of natural or semi-natural continuous habitats such as meadows and hedges of melliferous plants, untrampled areas, generalization of differentiated management of green spaces (intervention adapted to ecological issues according to the area), the banishment of the use of chemicals for the management of green spaces, reduction of mowing frequency and heights. These actions should be supported by raising the awareness of city-dwellers about the importance of the ecological management of green spaces for pollinators and why pollinators’conservation is essential for providing ecological services in cities.

Our results suggest that most of the main university campuses located in Seoul constitute suitable areas for pollinators due to their large areas of open spaces. Most Korean universities are private, and thus implement the green space management practices they prefer. As this is also the case for elementary, middle and high schools, engaging the education sector to integrate more green spaces could be a win-win situation. While improving habitat quality for pollinators, these spaces could be involved for educational purposes. In a city like Seoul where most young people are disconnected from nature, enhancing green spaces for biodiversity could increase opportunities to routinely experience nature [53]. In a country where stress linked to academic pressure leads to suicidal behavior [54,55,56], more experiences in nature would be a way to reduce stress and anxiety, thereby improving well-being.

Our results also implicitly illustrate where the main habitats are not. Some neighborhoods are drastically devoid of green spaces in Seoul such as the Gangnam district (see Figure 1, south-east of the river). The Yongsan Military Base, located in the center of Seoul (south of Namsan Mountain, see Figure 1) also lacks suitable habitats for flower visitors. These areas could also be involved in creating new habitats. The Bukhansan National Park (northern part of Seoul) was, surprisingly, not highlighted as suitable habitat for any of the main families according to our results. This might be due to the lack of observations gathered in this area and the lack of presence and access to flowering plants and shrubs. Another explanation is that this mountain, mainly constituted of rocky areas and pine tree forest leading to soil acidification that limit the presence of flowers, doesn’t constitute some of the best habitat for pollinator conservation. 

Most of the “Conservation corridors” are located in built-up areas, which means that the development of new green spaces will be limited in these areas. However, the city of Seoul has shown its capacity to recreate new green spaces by destroying previous highways in the city to reopen a stream (see Introduction section), and by converting railways or abandoned water treatment plants into public green spaces. Innovative landscaping and architectural approaches are crucial in order to integrate more suitable habitats for pollinators in the building arena, through the creation of green roofs and walls, where the interest in urban biodiversity is today well documented [57,58,59], even if some authors argued that these infrastructures may never be as suitable as ground-level habitats for biodiversity conservation [60]. At the street scale, urban tree bases are a way to promote plant dispersion and therefore constitute habitats for pollinators [61]; more generally, the vegetation of streets is providing ecological services as described by Säumel et al. [62]. However, the interest in these infrastructures would be reinforced by the planting of native plants and by the implementation of an ecological management approach. The best strategy would be to combine these actions with multi-functionality spaces for city-dwellers such as cycling and walking paths. Communication about more ecological management practices will also help users to better understand the issues of conservation in urban areas.

## 5. Conclusions

Citizen science is a powerful tool to enhance knowledge about biodiversity in areas such as Seoul where ecological data are lacking. As urbanization increases and cities become denser, it is crucial to promote sustainable urban planning, innovative architectural design, and management strategies that, (1) support ecosystem services provided by urban biodiversities, such as pollination, and (2) recreate opportunities for city-dwellers to interact with urban biodiversity. 

## Figures and Tables

**Figure 1 animals-12-01469-f001:**
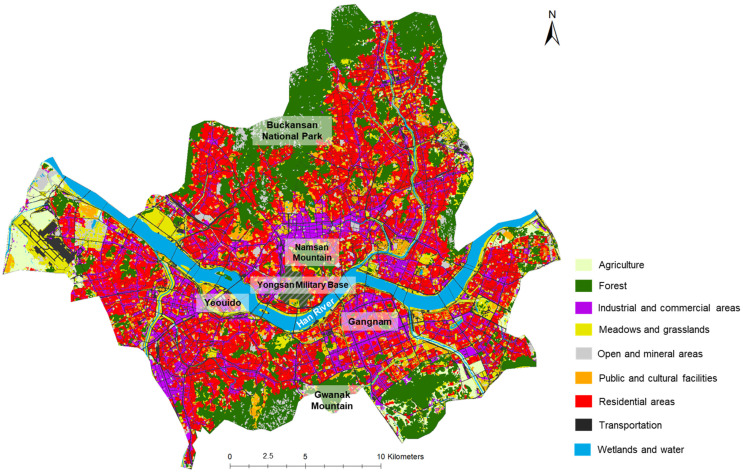
Land-use class of the city of Seoul (2017) and some major landmarks.

**Figure 2 animals-12-01469-f002:**
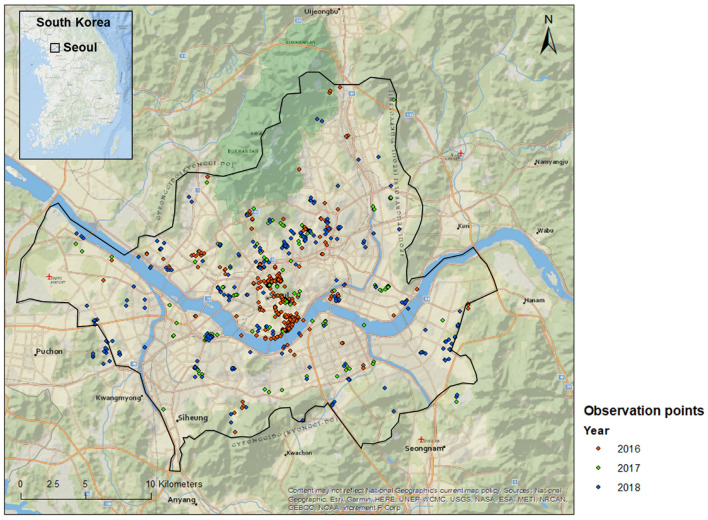
Locations of the observation points were mapped by the participants between 2016 and 2018.

**Figure 3 animals-12-01469-f003:**
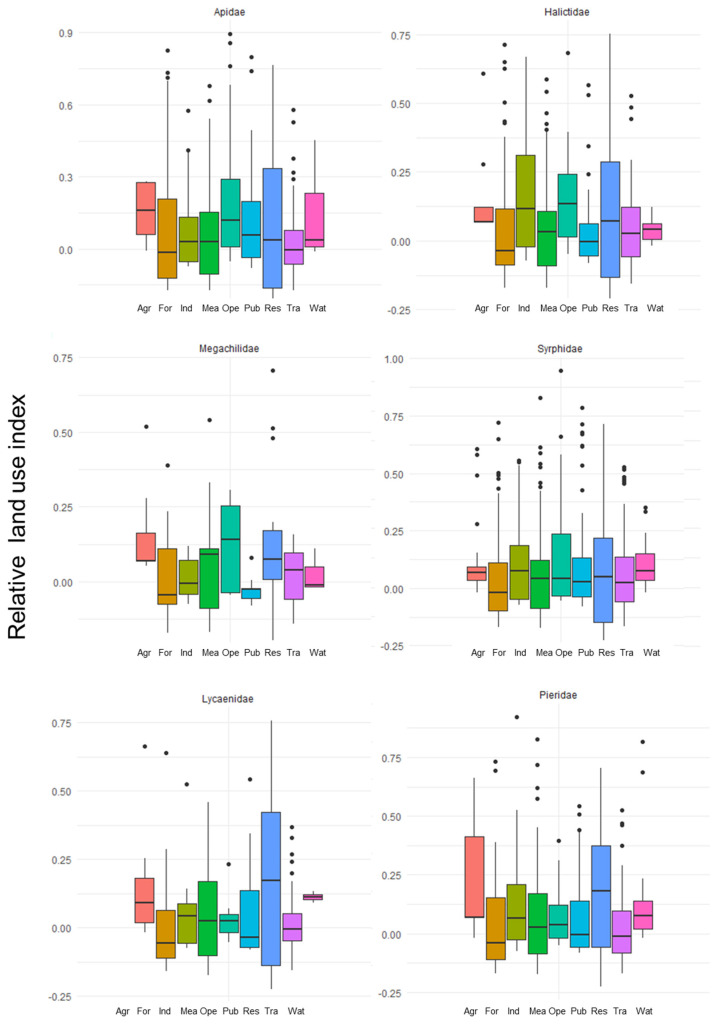
Relative land use affinity according to six land-use classes among the most frequent pollinators genus observed in Seoul between 2016 and 2018 (Agr: agriculture; For: forest; Ind: industrial and commercial areas; Mea: meadows and grasslands; Ope: open and mineral areas; Pub: public and cultural facilities; Res: residential areas; Tra: transportation, Wat: wetlands and water).

**Figure 4 animals-12-01469-f004:**
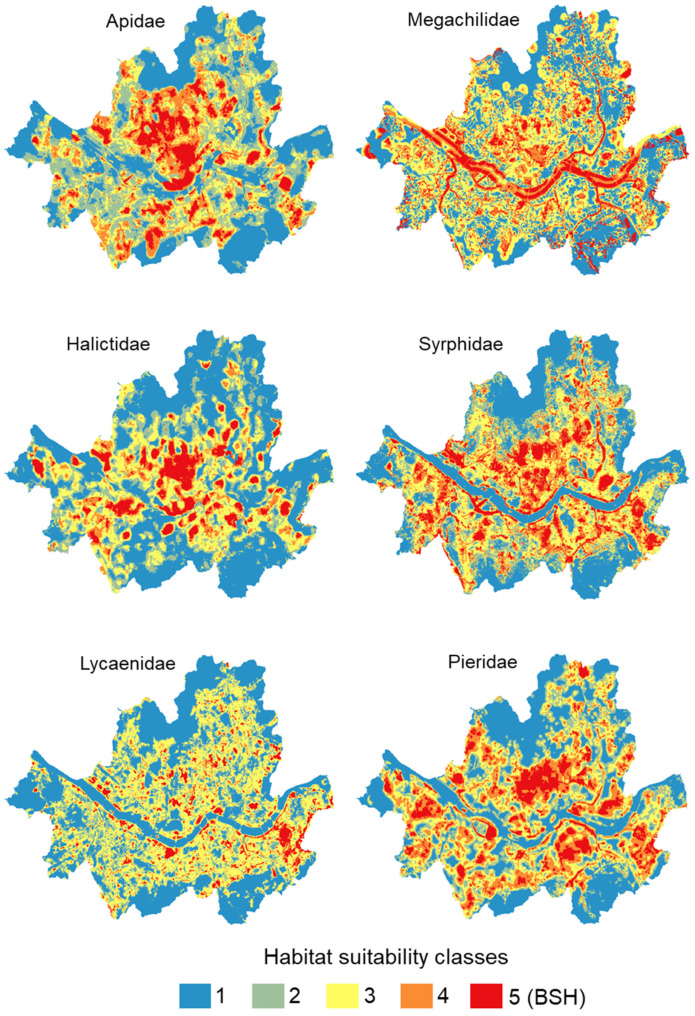
Estimated probability of observation of six families of pollinators in Seoul. For each family, the areas were ranked according to suitability classes ranging from 1 (blue areas, lowest probability of occurrence) to 5 (red areas, highest probability of occurrence, called in the manuscript “Best suitable habitats”, BSH).

**Figure 5 animals-12-01469-f005:**
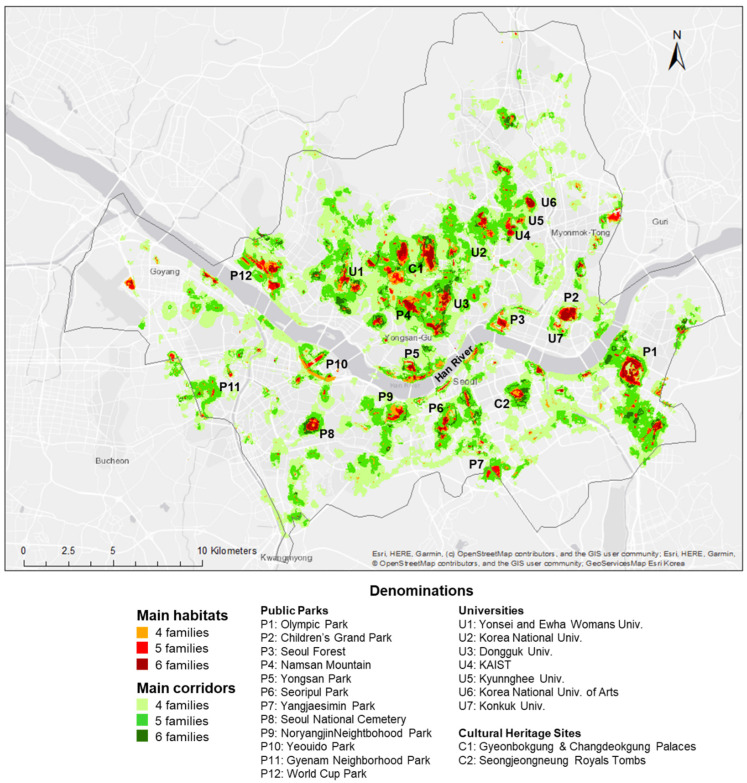
Map illustrating the main habitats and corridors for six families of pollinators in Seoul city. The main public parks, universities, and Cultural Heritage Sites are representing most of the largest main habitats.

**Figure 6 animals-12-01469-f006:**
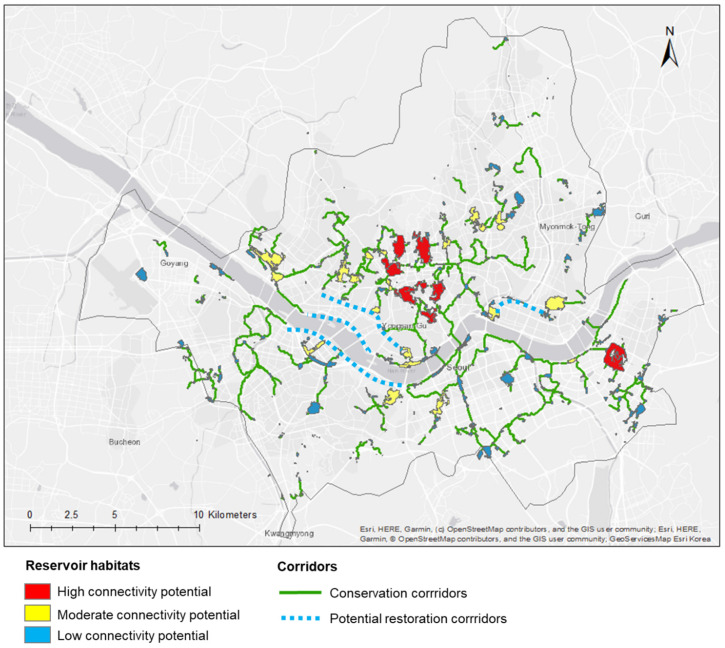
Synthesis map of corridors to maintain and restore to improve connectivity for wild pollinators in Seoul.

**Table 1 animals-12-01469-t001:** Parameters and statistics for selected Maxent models of pollinator families. True skill statistic (TSS) represents the combined true positive and negative rates, while the area under the curve (AUC) represents the overall model fit.

Family	Variables	Betamultiplier	Threshold	TSS	AUC
Apidae	13	2	0.4079	0.6047 ± 0.0936	0.8306 ± 0.0334
Halictidae	14	2	0.3635	0.6742 ± 0.1308	0.8706 ± 0.0330
Lycaenidae	53	2	0.4605	0.5860 ± 0.1181	0.9057 ± 0.0547
Megachilidae	9	3	0.4785	0.5302 ± 0.1867	0.8624 ± 0.0563
Pieridae	11	2	0.4529	0.5496 ± 0.0816	0.8276 ± 0.0411
Syrphidae	15	2	0.4276	0.5726 ± 0.0554	0.8272 ± 0.0324

**Table 2 animals-12-01469-t002:** Total pollinator occurrences recorded by citizen scientists between 2016 and 2018. The number of occurrences is given for each order, super-family, and family as well as the percentage of observations they represent.

Classification	Total Occurrences	Percentage of Observations
Hymenoptera	1061	57.6%
Apoidae	901	48.9%
Apidae	684	37.1%
Halictidae	147	8.0%
Megachilidae	37	2.0%
Andrenidae	10	0.5%
Crabronidae	6	0.3%
Colletidae	2	0.1%
Other or not identifiable at the family scale	15	0.8%
Formicoidae	54	2.9%
Vespoidae	37	2.0%
Tenthredinoidae	8	0.4%
Chrysidoidea	2	0.1%
Ichneumonoidae	2	0.1%
Other/not identifiable at the super-family scale	57	3.1%
Diptera	326	17.7%
Syrphidae	225	12.2%
Muscomorpha	30	1.6%
Sarcophagoidae	17	0.9%
Other or not identifiable at the super-family scale	54	2.9%
Lepidoptera	288	15.6%
Papilionoidae	248	13.5%
Lycaenidae	74	4.0%
Nymphalidae	22	1.2%
Papilionidae	10	0.5%
Pieridae	142	7.7%
Hesperioidea	13	0.7%
Bombycoidae	5	0.3%
Zygaenidae	2	0.1%
Other or not identifiable at the super-family scale	18	1.0%
Hemiptera	105	5.7%
Lygaeoidea	48	2.6%
Coreoidae	18	1.0%
Pentatomoidae	13	0.7%
Other or not identifiable at the super-family scale	26	1.4%
Coleoptera	49	2.7%
Scarabaeoidae	10	0.5%
Chrysomeloidae	8	0.4%
Cucujoidae	7	0.4%
Curculionoidae	5	0.3%
Other or not identifiable at the super-family scale	19	1.0%
Othoptera	6	0.3%
Arachnida	6	0.3%
Thomisidae	5	0.3%
Other or not identifiable at the family scale	1	0.1%
Odonata	1	0.1%

**Table 3 animals-12-01469-t003:** Observed and estimated species richness (number of genera) by land-use classes.

Land Class	Observed Richness	Estimated Jackknife	Estimated Bootstrap
Agriculture	53	83.7 ± 5.5	66.2 ± 2.6
Forest	44	75.7 ± 5.6	55.9 ± 2.5
Industry & Commercial	29	47.7 ± 4.3	36.7 ± 1.2
Meadows and grasslands	61	92.8 ± 5.6	74.5 ± 2.8
Open and mineral areas	30	47.7 ± 4.2	37.1 ± 1.9
Public and cultural facilities	37	61.7 ± 4.9	46.9 ± 2.3
Residential areas	52	79.9 ± 5.3	63.3 ± 2.6
Transportation	34	51.9 ± 4.2	41.1 ± 2.1
Wetlands and water	9	14.7 ± 2.3	11.2 ± 1.1

## Data Availability

Data sources (mainly Serret et al., 2019) are cited in the Introduction and Methods sections of the manuscript text.

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
