# Peer review of "Towards Ecological Management and Sustainable Urban Planning in Seoul, South Korea: Mapping Wild Pollinator Habitat Preferences and Corridors Using Citizen Science Data"

_animals, 2022, doi:10.3390/ani12111469_

Round 1

Reviewer 1 Report

I think this is well-written study without any major flaws. However, I feel while reading the paper that the imperfect original data is stretched too far with fancy statistics. I also wonder how similar results one would have been obtained just with observing green spaces from the city map.

Other more detailed comments are:

L156-160. This list of land use categories is not similar as in the Fig. 1. Use the same terms in both. This applies also to Table 3 and Fig. 3 (rocks-open areas-naked lands –ope). Try to use these also in the same order every occasion. Also, Fig. 3 needs a legend explaining the land use categories (= colours)

L161-164. Were there observations in every land use category? It is impossible to make extrapolations if the basic information is non-existent.

2.5.1. Could you clarify what are the 52 variables. I have troubles to end up this number.

2.5. In my opinion this part of habitat suitability modeling is not adequately explained. It is impossible to follow if you are not a specialist on SDMs, and probably difficult even for specialist.

L241: There is no section 2.4.3. Should be 2.5.3.?

Table 3. Legend. (number of species) or (number of genera)?

L334-339. For example, if you have generally less observations in some habitat type, does this automatically lead to less affinity to this habitat? For example, is the poor affinity to forests because the focal species avoid forests or due to fact that people did not make observations in the forests. In more general, how the distribution of observations in relation to different habitats affect your results. Maybe this should be included in the discussion, if the authors feel that this is an important point affecting the results.

L345-383. Names of parks etc. are not interest of other readers than those from Seoul.

L422-424. Even the most sophisticated statistical methods cannot generate data from the scratch.

L439-440. Could you be more specific what kinds of management actions are needed?

Discussion. In my mind the discussion is too Seoul-centric for wider audience. Also, discussion is not focused at all on interesting ecological differences observed between species groups in their probability areas (Fig. 4), or other pure ecological findings.

Author Response

Reviewer 1:

I think this is well-written study without any major flaws. However, I feel while reading the paper that the imperfect original data is stretched too far with fancy statistics. I also wonder how similar results one would have been obtained just with observing green spaces from the city map.

>> We agree that data from citizen sciences may content some bias linked to unbalanced and not standardized sampling design. However, we think that our paper allows to hierarchize the importance of these different green spaces for the wild pollinators, backed up by modeling and statistical analysis that is not possible with only observation.

Other more detailed comments are:

L156-160. This list of land use categories is not similar as in the Fig. 1. Use the same terms in both. This applies also to Table 3 and Fig. 3 (rocks-open areas-naked lands –ope). Try to use these also in the same order every occasion.

>> Changes have been done. The land-use class are organized following alphabetic order :  “Land-use was categorized into nine classes: agriculture, forest, industrial and commercial areas, meadows and grasslands (including planted lawns, planted or natural meadows), open and mineral areas, public and cultural facilities, residential areas (mainly residential towers with small green spaces or lower buildings), transportation, wetlands and water (Figure 1).”

Also, Fig. 3 needs a legend explaining the land use categories (= colours)

>> Legend has been modified: “Figure 3. Relative land use affinity according to six land use classes among the most frequent pollinators genus observed in Seoul between 2016 and 2018 (Agr : agriculture ; For : forest : Ind : industrial and commercial areas ; Mea : meadows and grasslands ; Ope : open and mineral areas ; Pub : public and cultural facilities ; Res : residential areas ; Tra : transportation, Wat : wetlands and water).”

L161-164. Were there observations in every land use category? It is impossible to make extrapolations if the basic information is non-existent.

>> We agree that some of the land uses such as forests have been less observed than other types of habitats. However, we considered that the several observations we had in these types of under-represented habitats were sufficient to conduct our analysis. The methodology we used from Palmer allows extrapolations even if the sampling is low (see Palmer 1990)

2.5.1. Could you clarify what are the 52 variables. I have troubles to end up this number.

>> We have added explanations in parentheses to help the reader follow the number of variables:

“(total seven variables)” Line 230

“(total nine variables)” Line 232

“(total 36 variables; four each of nine classes)” Line 233

2.5. In my opinion this part of habitat suitability modeling is not adequately explained. It is impossible to follow if you are not a specialist on SDMs, and probably difficult even for specialist.

>> Maximum entropy modeling is the most commonly used method in species distribution modeling. Besides adding more explanation specific to the package used for modeling (below), it is difficult for us to know what is needed to further explain this section, as all of the information relevant to recreating the modeling is already provided.

“This package runs MaxEnt models for multiple runs, removing variables that are highly correlated and do not contribute greatly to the models. This package can be run with different betamultiplier values representing a “smoothing” of a species’ response to each model variable.” (Line 237-240)

We have also added some explanation as to what MaxEnt entails:

“Maximum entropy (MaxEnt [39]) modeling is one such method for predicting suitability based on a species’ response to each environmental variable compared to background conditions” (Line 216-218)

L241: There is no section 2.4.3. Should be 2.5.3.?

>> We have corrected this in the revised manuscript

Table 3. Legend. (number of species) or (number of genera)?

>> Yes, this was a mistake, the title has been changed into “number of genera”

L334-339. For example, if you have generally less observations in some habitat type, does this automatically lead to less affinity to this habitat? For example, is the poor affinity to forests because the focal species avoid forests or due to fact that people did not make observations in the forests. In more general, how the distribution of observations in relation to different habitats affect your results. Maybe this should be included in the discussion, if the authors feel that this is an important point affecting the results.

>> We agree that data collection based on citizen science project are often lacking of balanced sampling, leading to sometimes some biais in the results. We are totally aware of this limit and will it make clearer in the discussion

L345-383. Names of parks etc. are not interest of other readers than those from Seoul.

>> We thought it would be important to mention the parks names as we located them on another maps (Figure 5).

L422-424. Even the most sophisticated statistical methods cannot generate data from the scratch.

>> We agree and are aware that this is the limit of our work that we exposed in the discussion:

“As is the case for most citizen science programs, the sampling was affected by the location where people took pictures. Due to a biased sampling design, we also lacked suffi-cient information from specific types of habitats such as forests, private backyards that are not common in a city like Seoul and restrictive access areas, such as the Yongsan military base. Even if these issues can nevertheless be addressed by statistical solutions [27,51], more data would be needed to strengthen the results and predictions of this work  and further field campaigns (or call to observers) targeting under-represented habitats could address current sampling limitations.”

L439-440. Could you be more specific what kinds of management actions are needed?

>> We added a part in the discussion : Conservation and ecological management actions to enhance habitats for pollinators could be done at different levels: conservation, restauration and creation of natural or semi-natural continuous habitats such as meadows and hedges of melliferous plants, un-trampled areas, generalization of differentiated management of green spaces (intervention adapted to ecological issues according to the area), banishment of use of chemicals for the management or green spaces, reduction of mowing frequencies and heights. These actions should be supported by raising city-dwellers awareness’s about the importance of ecological management of green spaces for pollinators and why pollinators ‘conservation is essentials for providing ecological services in cities.”

Discussion. In my mind the discussion is too Seoul-centric for wider audience. Also, discussion is not focused at all on interesting ecological differences observed between species groups in their probability areas (Fig. 4), or other pure ecological findings.”

>> We added a paragraph in the discussion about these results: “The results based on land-use affinities showed that agricultural areas were favourable to all the groups of pollinators, showing that these areas were a way to create suitable habitats for pollinators while addressing provisioning services in the city of Seoul. Transportation areas may also be a way to create habitats for pollinators and wildlife in general. Indeed, the surroundings of the tracks are often less managed and are presenting small patches of wastelands with various species of plants. All of these areas could be better consider in order to create a network of suitable habitats for pollinators.”

Reviewer 2 Report

The introduction describes the problem quite well but I miss a passage about pollinator habitat in general and characteristics of corridors for insects. Wildlife is a very general term and the paper is especially about pollinators, this should be considered in the introduction.

Material and methods are well described only the link www.egis.me.kr is not working.

Table 2 – I guess with occurrences you refer to the 250 m squares described in 2.3. If yes, please describe it. Otherwise the difference between occurrence and observation is not clear.

4.1 I prefer a more detailed discussion about the number of species found compared to the number of species expected, especially I would like to see more numbers and not a more qualitative, general discussion. You can do it also on the family level.

I am wondering about the lack of suitable habitats (Fig. 5 and 6) in the north of Seoul in comparison to Fig. 4 where you find high BSH for several families. Even if you discuss the lack of observations (line 458) in this area, a model for habitat suitability of connectivity should compensate this lack of observations. As you already discussed flowering plants are the key for pollinator presence and in general in woodland areas you got lower numbers of flowering plants because of the lack of light. I a wondering that a military training ground shows low suitability. Normally there is a lot of disturbance vegetation and soil with leads to a high diversity in animals and plants.

I agree that such projects can be a very useful tool for urban planning but you should use models to show the future or the potential too and not only the presence.

Author Response

The introduction describes the problem quite well but I miss a passage about pollinator habitat in general and characteristics of corridors for insects. Wildlife is a very general term and the paper is especially about pollinators, this should be considered in the introduction.

>> A paragraph into the introduction were completed: “Among species present in urban areas, wild pollinators are important for maintain-ing urban ecosystems and can be considered as bioindicators of environmental health [16,17]. Pollination is indeed one of the most important ecosystem services and is crucial for the persistence of diverse plant communities [18] and to insure the productivity of urban agriculture [19]. Researchers have only recently realized that cities could be designed as “refuges” for wild pollinators thanks to areas creating suitable and continuous habitats such as parks, residential gardens or green spaces a business sites. Indeed, some of these urban green spaces are hosting melliferous flowers, grasslands, hedges and other favora-ble micro-habitats for pollinators, creating a network of nesting areas within cities [20,21]. To this end, there is a need to better understand ecological requirements of pollinator communities within cities, highlighting habitat preferences and identifying potential corridors in the city as first steps to implement conservation actions [22].”

Material and methods are well described only the link www.egis.me.kr is not working.

>> The link has been updated : https://egis.me.go.kr/

Table 2 – I guess with occurrences you refer to the 250 m squares described in 2.3. If yes, please describe it. Otherwise the difference between occurrence and observation is not clear.

>> The table 2 presents the total of the observations that have been made. We made it clearer in the results: “In total, 178 species belonging to 128 genera and 60 families were observed before the fishnet was applied (Table 2).” And also in the legend: “Total pollinators’ occurrences recorded by citizen scientists between 2016 and 2018.”

4.1 I prefer a more detailed discussion about the number of species found compared to the number of species expected, especially I would like to see more numbers and not a more qualitative, general discussion. You can do it also on the family level.

>> We made it clearer in the discussion: “Thanks to the pollinator data collected by citizen scientists between 2016 and 2018, we were able to identify a total 178 wild pollinators species.”

“For the city of Seoul, our study is the first investigating the diversity of pollinators. Our study showed that the most represented family was the Hymenopterans, meaning that the pollination service within Seoul is more dependent on this type of family, fol-lowed by Dipterans and Lepidopterans. Knowing that fact, it is possible to adapt conser-vation and ecological restoration measures targeting these families of pollinators.”

I am wondering about the lack of suitable habitats (Fig. 5 and 6) in the north of Seoul in comparison to Fig. 4 where you find high BSH for several families.

>> The Fig. 5 has been done with the addition of the different layers showed Fig. 4 and we retrain only patches with at least 4 superposed layers (habitat for 4 families).

Even if you discuss the lack of observations (line 458) in this area, a model for habitat suitability of connectivity should compensate this lack of observations. As you already discussed flowering plants are the key for pollinator presence and in general in woodland areas you got lower numbers of flowering plants because of the lack of light. I a wondering that a military training ground shows low suitability. Normally there is a lot of disturbance vegetation and soil with leads to a high diversity in animals and plants.

>> It is hard to tell as we have no access to the land-uses of these areas. This military based is closer from residential areas (with some trees and open weadows and grassland) than a training area.

I agree that such projects can be a very useful tool for urban planning but you should use models to show the future or the potential too and not only the presence.

>> That is what we tried to show thanks to the modelling of the potential connectivity and corridors that could be enhance.

Round 2

Reviewer 1 Report

Your response and corrections made are satisfying.